# High KDM1A Expression Associated with Decreased CD8+T Cells Reduces the Breast Cancer Survival Rate in Patients with Breast Cancer

**DOI:** 10.3390/jcm10051112

**Published:** 2021-03-07

**Authors:** Hyung Suk Kim, Byoung Kwan Son, Mi Jung Kwon, Dong-Hoon Kim, Kyueng-Whan Min

**Affiliations:** 1Department of Surgery, Division of Breast Surgery, Hanyang University Guri Hospital, Hanyang University College of Medicine, Guri 11923, Korea; hyung6960@naver.com; 2Department of Internal Medicine, Eulji Hospital, Eulji University School of Medicine, Seoul 03181, Korea; sbk1026@eulji.ac.kr; 3Department of Pathology, Hallym University Sacred Heart Hospital, Hallym University College of Medicine, Anyang 14068, Korea; mulank@hanmail.net; 4Department of Pathology, Kangbuk Samsung Hospital, Sungkyunkwan University School of Medicine, Seoul 03181, Korea; 5Department of Pathology, Hanyang University Guri Hospital, Hanyang University College of Medicine, Guri 11923, Korea

**Keywords:** KDM1A, tumor infiltrating lymphocyte, breast neoplasm, prognosis

## Abstract

Background: Lysine-specific demethylase 1A (KDM1A) plays an important role in epigenetic regulation in malignant tumors and promotes cancer invasion and metastasis by blocking the immune response and suppressing cancer surveillance activities. The aim of this study was to analyze survival, genetic interaction networks and anticancer immune responses in breast cancer patients with high KDM1A expression and to explore candidate target drugs. Methods: We investigated clinicopathologic parameters, specific gene sets, immunologic relevance, pathway-based networks and in vitro drug response according to KDM1A expression in 456 and 789 breast cancer patients from the Hanyang university Guri Hospital (HYGH) and The Cancer Genome Atlas, respectively. Results: High KDM1A expression was associated with a low survival rate in patients with breast cancer. In analyses of immunologic gene sets, high KDM1A expression correlated with low immune responses. In silico flow cytometry results revealed low abundances of CD8+T cells and high programmed death-ligand 1 (PD-L1) expression in those with high KDM1A expression. High KDM1A expression was associated with a decrease in the anticancer immune response in breast cancer. In pathway-based networks, KDM1A was linked directly to pathways related to the androgen receptor signaling pathway and indirectly to the immune pathway and cell cycle. We found that alisertib effectively inhibited breast cancer cell lines with high KDM1A expression. Conclusions: Strategies utilizing KDM1A may contribute to better clinical management/research for patients with breast cancer.

## 1. Introduction

Breast cancer is a heterogeneous group of cancers that occur in breast tissues, and the incidence is rapidly increasing worldwide. Breast cancer is the most common cancer among females [1]. Recently, the number of early diagnoses of breast cancer by screening has risen, and the survival rate of breast cancer patients has increased as a result of the implementation of systemic adjuvant treatment. However, breast cancer is still the second leading cause of cancer mortality among women in the United States, and the decline in breast cancer mortality has slowed slightly in recent years [2]. To improve the survival of patients with breast cancer, it is important to understand the molecular signaling pathways and biomarkers associated with breast cancer development, progression and metastasis and to discover molecular targeted agents for clinical applications.

Epigenetic modification plays an important role in gene expression and physiological development as well as various disease pathogeneses. It also plays a pivotal role in the development and progression of different types of malignancies, such as breast cancer [3,4]. Methylation is a major type of histone modification and has a critical role in epigenetic modification of gene expression. Histone methylation primarily acts on lysine residues to regulate the suppression or activation of gene expression. Specific lysine methyltransferases (KMTs) and lysine demethylases (KDMs) reversibly regulate histone lysine methylation and are associated with several biological processes and diseases in humans [5,6,7].

KDM1A, also named lysine-specific histone demethylase 1 (LSD1), is a flavin-dependent histone demethylase of mono- and dimethylated histone H3 lysine 4 (H3K4me1/2) that was first reported by Shi, Y et al. [8]. KDM1A exhibits the ability to demethylate H3K4me1/2 or H3K9me1/2 to repress or activate transcription and mediate progression of some cancers by demethylating nonhistone proteins [9,10,11]. High KDM1A expression plays a key role in tumorigenesis since it removes methyl groups from methylated histone H3 lysine 4 (H3K4) and H3 lysine 9 (H3K9), thereby increasing cell invasiveness, motility and proliferation and inhibiting normal cell differentiation through chromatin modification [12,13,14]. Previous studies have demonstrated that high KDM1A expression is related to worse prognoses in various types of solid tumors, such as nonsmall cell lung cancer, colorectal cancer, neuroblastoma, prostate cancer and breast cancer [15,16,17,18,19]. Other studies have reported that KDM1A is highly expressed in ER-negative breast cancer, with a decreased survival rate. Furthermore, it has been shown that high KDM1A expression is associated with progression from ductal carcinoma in situ (DCIS) to invasive ductal carcinoma (IDC) in breast cancer [19,20]. Nevertheless, the molecular mechanisms of carcinogenesis and clinicopathological differences according to KDM1A expression in breast cancer have not yet been completely explained.

The aim of our study was to evaluate the degree of KDM1A expression in patients with breast cancer and to analyze the relationship between clinicopathological parameters and survival rate versus KDM1A expression in a cohort from Hanyang university Guri Hospital (HYGH) and one from The Cancer Genome Atlas (TCGA) database [21]. We also investigated gene sets related to KDM1A using gene set enhancement analysis (GSEA) and a pathway-based network [22,23,24]. Anticancer immune responses according to KDM1A expression were analyzed through distributions of tumor-infiltrating lymphocytes such as CD8+T cells and CD4+T cells. By performing high-throughput drug sensitivity screening using the Genomics of Drug Sensitivity in Cancer (GDSC) database, we also identified new drug candidates for breast cancer with high KDM1A expression [25,26] (Figure 1).

## 2. Materials and Methods

### 2.1. Patient Selection

We identified 456 patients with IDC of the breast who underwent surgery at HYGH in Korea between 2005 and 2015. The Reporting Recommendations for Tumor Marker Prognostic Studies (REMARK) criteria were followed throughout our study [27]. All eligible patients had histopathological evidence of primary IDC confirmed by pathologists using core needle biopsy of the breast, and clinicopathological results could be confirmed through medical records. Patients for whom paraffin blocks were unavailable or clinicopathological factors, including clinical outcomes, were incomplete were excluded. We collected data on tumor and nodal stage, histopathological grade and lymphovascular and perineural invasion. We obtained the survival rate using data from our cohort and TCGA. Disease-free survival (DFS) was defined as survival from the date of diagnosis to recurrence/new distant metastasis, and disease-specific survival (DSS) was defined as survival from the date of diagnosis to cancer-related death. This study protocol was approved by the Institutional Review Board of Hanyang university Guri Hospital (IRB number: 2020-04-015) and conducted in accordance with the Declaration of Helsinki.

### 2.2. Tissue Microarray Construction and Immunohistochemistry

Tissue microarray (TMA) blocks were assembled using a tissue array instrument (AccuMax Array; ISU ABXIS Co., Ltd., Seoul, Korea). We used duplicate 3-mm-diameter tissue cores (tumor components in a tissue core > 70%) from each donor block. Four-micrometer sections were cut from the TMA blocks using routine techniques. Immunostaining for KDM1A (1:300, Novus Biologicals, Littleton, CO, USA), estrogen receptor (ER) (1:200, Lab Vision Corporation, Fremont, CA, USA), progesterone receptor (PR) (1:200, Dako, Glostrup, Denmark), human epidermal growth factor receptor 2 (HER2) (1:1, Ventana Medical Systems, Tucson, AZ, USA), P53 (1:5000, Cell Marque, Hot Springs, AR, USA) and Ki67 (1:200; MIB-1, Dako, Glostrup, Denmark) was performed using Bond Polymer Refine Detection System (Leica Biosystems Newcastle Ltd., Newcastle, UK) according to the manufacturer’s instructions. Anti-PD-L1 (clone SP142, Ventana Medical Systems, Roche, Tucson, AZ, USA) staining was performed. Immunohistochemical staining (IHC score) was evaluated using the semiquantitative Remmele scoring system [28], which links the IHC staining intensity (SI) with the percentage of positive cells (PP) (Figure 2A). To determine the optimal cutoff values of KDM1A, receiver operating characteristic (ROC) curves plotting sensitivity versus 1—specificity were used. The cutoff value calculated by the ROC was used to evaluate the relationship between cancer specific death events and KDM1A expression. High KDM1A expression was defined as ≥5.

### 2.3. Gene Sets, In Silico Cytometry and Network Analysis from TCGA

We obtained 789 IDC cases with RNA-Seq from TCGA [29]. The RNA seq from TCGA was calculated, and KDM1A expression was determined as either low (log2 -transformed scores < 11.543) or high (log2 -transformed scores > 11.543). We analyzed significant gene sets using gene set enrichment analysis (GSEA, version 4.03, from the Broad Institute at MIT and Harvard) from the Broad Institute at MIT. The gene set (4872 immunologic signatures) was used to identify gene sets associated with high KDM1A expression [22]. For this analysis, 1000 permutations were used to calculate *p*-values, and the permutation type was set as follows: *p* < 0.05, false discovery rate (FDR) of <0.2 and family wise-error rate (FWER) of ≤0.4. We applied in silico flow cytometry to determine the proportions of 22 subsets of immune cells using 547 genes [30]. Pathway-based network analyses were visualized using Cytoscape software. To interpret the immunologic relevance of KDM1A and its relevant elements in IDC, we performed functional enrichment analysis to clarify functionally grouped gene ontology, and pathway annotation networks were visualized using Cytoscape (version 3.8) [23,24].

### 2.4. Data Extraction from the GDSC Database

We analyzed the relationship between anticancer drug sensitivity and KDM1A expression based on the Genomics of Drug Sensitivity in Cancer (GDSC version 2) dataset, which contains drug response data for approximately 50 breast cancer cell lines to 175 anticancer drugs [31]. We measured anticancer drug sensitivity in 50 breast cancer cell lines with the natural log-half-maximal inhibitory concentration (LN IC50). The drug response was defined as the LN IC50. A drug was identified as effective when the calculated LN IC50 value was decreased in cell lines with high KDM1A expression and increased in those with low KDM1A expression, i.e., when an inverse correlation was observed. Pearson’s correlation and Student’s *t*-test between LN IC50 values and KDM1A expression were performed [25,26].

### 2.5. Statistical Analysis

Correlations between clinicopathological parameters and KDM1A were analyzed using the χ^2^ test and a linear-by-linear association test. Student’s *t*-test and Pearson’s correlation were applied to examine differences among continuous variables. Survival rates were compared using the log-rank test and Cox regression analyses. A two-tailed *p*-value of <0.05 was considered statistically significant. All data were analyzed using R packages and SPSS statistics (version 25.0; SPSS Inc., Chicago, IL, USA).

## 3. Results

### 3.1. Clinicopathologic Characteristics of KDM1A Expression in Breast Cancer

A total of 1245 patients with KDM1A expression data and survival data in the HYGH cohort and TCGA were divided into a high expression group and a low expression group using the optimal cutoff. Among 456 patients in the HYGH group, KDM1A was highly expressed in 210 (46.1%). Comparison of the detailed clinicopathologic characteristics of patients with high or low KDM1A expression is described in Table 1. High KDM1A expression was associated with higher pathologic nodal (N) stage (*p* = 0.001), higher histological grade (*p* < 0.001), and more lymphatic invasion (*p* = 0.004) and p53 (*p* < 0.001) compared to low KDM1A expression. High KDM1A expression was also related to ER negativity (*p* = 0.035), PR negativity (*p* = 0.020), and HER2 receptor positivity (*p* < 0.001). Based on TCGA data, high KDM1A expression and increased copy number variation were observed in IDC tissue. High KDM1A expression was associated with hypomethylation and IDC compared to normal breast tissue (*p* < 0.001) (Figure 2B).

### 3.2. Prognostic Value of KDM1A Expression in Breast Cancer

We analyzed the survival rate of 456 patients in the HYGH group according to KDM1A expression status. Patients with high KDM1A expression had significantly worse DFS and DSS than those with low KDM1A expression (all *p* < 0.001) (Figure 2C). Multivariate analyses were performed to investigate prognostic factors affecting DFS and DSS in the HYGH group, and high KDM1A expression was identified as an independent prognostic factor for DFS and DSS (HR = 3.241; 95% CI, 1.969–5.336, *p* < 0.001 and HR = 4.095; 95% CI, 2.477–6.770, *p* < 0.001, respectively) (Table 2). We investigated 789 IDC patients in TCGA to confirm the reliability of our results regarding the significant relationship between KDM1A and survival; compared to low KDM1A expression, high KDM1A expression correlated significantly with poor DFS and DSS (*p* = 0.039 and *p* = 0.035, respectively) (Figure 2D).

### 3.3. Anticancer Immune Response, Gene Set Enrichment Analysis and Pathway Network Analysis of KDM1A

We analyzed the relationship between KDM1A expression and immunity using the HYGH and TCGA cohorts. In HYGH, our results showed a significant correlation between high KDM1A expression and a high tumor-infiltrating lymphocyte (TIL) percentage (*p* = 0.004). Among them, high expression of KDM1A showed a significant correlation with low CD8+T cell count and high CD4+T cell count (*p* = 0.001 and 0.004, respectively). Additionally, high KDM1A expression showed a significant correlation with high programmed death-ligand 1 (PD-L1) expression (*p* = 0.001) (Figure 3A). In TCGA, a high KDM1A mRNA level correlated significantly with a high TIL percentage, low fraction of CD8+T cells, high fraction of CD4+T cells and high level of cancer/testis antigens (CTAs) (*p* = 0.039, 0.038, 0.012, 0.009, respectively) (Figure 3B).

We identified a variety of CD8+T cell downregulation gene sets associated with high KDM1A expression through GSEA and as a result identified 10 significant gene sets (GSE39110 DAY3 vs. DAY6 POST IMMUNIZATION CD8 TCELL DN, GSE10239 NAIVE vs. KLRG1HIGH EFF CD8 TCELL DN, GSE39110 UNTREATED vs. IL2 TREATED CD8 TCELL DAY3 POST IMMUNIZATION DN, GSE10239 NAIVE vs. KLRG1INT EFF CD8 TCELL DN, GSE15930 NAIVE vs. 24H IN VITRO STIM CD8 TCELL DN, GSE15930 NAIVE vs. 24H IN VITRO STIM IL12 CD8 TCELL DN, GSE10239 NAIVE vs. DAY4.5 EFF CD8 TCELL DN, GSE19825 CD24LOW vs. IL2RA HIGH DAY3 EFF CD8 TCELL DN, GSE30962 ACUTE vs. CHRONIC LCMV PRIMARY INF CD8 TCELL DN, GSE15930 NAIVE vs. 48H IN VITRO STIM IFNAB CD8 TCELL DN) (Figure 4) (Appendix A). Moreover, molecular interaction pathway network analysis linked KDM1A directly to the androgen receptor (AR) signaling pathway and indirectly to the cell cycle, miRNA regulation of DNA damage response, ATM signaling network in development and disease, DNA damage response, ATM signaling pathway, DNA IR-damage and cellular response via ATR, DNA IR-double-strand break cellular response via ATM, integrated cancer pathway, integrated breast cancer pathway, regulation of megakaryocyte differentiation, somatic diversification of immune receptors via germline recombination within a single locus, regulation of B cell proliferation, and somatic recombination of immunoglobulin gene segments (Figure 5).

### 3.4. Drug Screening of Breast Cancer Cell Lines in the GDSC Database

Using the LN IC50 standard values presented in the GDSC database, we analyzed the drug sensitivity of 50 breast cancer cell lines according to KDM1A expression (Appendix A). Our results defined an effective target drug for KDM1A as one that shows a significantly high negative correlation between KDM1A expression and LN IC50 in Pearson correlation analysis As a result, we identified the following 6 anticancer drugs as those that can most effectively reduce the growth of breast cancer cells with high KDM1A expression: alisertib, MK-2206, dinaciclib, vorinostat, camptothecin, and entinostat (Figure 6). Among the six effective anticancer drugs, alsertib showed the most significant result of a highly negative correlation between high KDM1A expression and LN IC50, as based on Pearson correlation.

## 4. Discussion

Breast cancer mortality has declined significantly due to increased screening and the implementation of systemic adjuvant treatment in the past 15 years; nonetheless, approximately 25% to 40% of breast cancer patients develop metastasis, the leading cause of death [1]. Thus, research on the molecular mechanisms and signaling pathways involved in breast cancer metastasis and progression is important to improve survival. Furthermore, epigenetic regulation via methylation plays an important role that is closely related to the development and progression of breast cancer [5,6,7]. In addition, methylation status can provide critical information for advanced therapeutic trials in some patients [32].

Our study demonstrated that high KDM1A expression is significantly associated with worse clinicopathological parameters in patients with breast cancer, with high KDM1A expression indicating a higher frequency of lymph node metastasis, higher histological grade, and more lymphatic invasion. Survival analysis showed that compared to low KDM1A expression, high KDM1A expression is related to poor DFS and DSS in patients with breast cancer. To further confirm the implications of our results, the correlation between survival data from TCGA, a large-scale database, and KDM1A was analyzed, and the same results were obtained. We suggest that high expression of KDM1A may play critical oncogenic roles in breast cancer.

KDM1A participates in the regulation of gene expression by removing methyl groups from monomethylated and dimethylated lysine 4 of histone H3 and lysine 9 of histone H3. KDM1A plays critical roles as an important regulator of various aspects of cancer, such as oncogenes, which promote cancer growth, progression and metastasis [33,34]. A previous report found that KDM1A expression differs significantly between ductal carcinoma in situ and invasive ductal carcinoma of the breast. KDM1A expression was also found to be significantly higher in high-grade ductal carcinoma in situ than in low-grade ductal carcinoma in situ [20]. High KDM1A expression is related to ER-negative breast cancer as well as a worse prognosis [19]. Our study revealed higher KDM1A expression in breast cancer tissue than in normal tissue. High KDM1A expression was associated with aggressive clinical outcomes, such as lymph node involvement, high histological grade, HER2 positivity and lymphatic invasion, which can be found in studies of other cancers [35]. However, the specific mechanism of how high KDM1A expression affects breast cancer remains poorly understood.

Using GSEA of KDM1A and immunologic gene sets, we found relationships between high KDM1A expression and downregulation of CD8+T cell gene sets. One of the best predictors of the immune response to eliminate cancer cells is the number and phenotype of CD8+T cells recruited to the tumor site [36]. According to immunohistochemistry of the HYGH group, high KDM1A expression correlated with a high TIL percentage. High KDM1A expression was also related to decreased CD8+T cell abundance and upregulated PD-L1 expression. Increased TIL is known to be a good prognostic factor for cancer because lymphocytes play an important role in anti-cancer immunity. Among lymphocytes, CD8+T cell play the most important role in anti-cancer immunity. Our results showed that high expression of KDM1A was associated with an increase in TIL, while CD8+T cell, which play an important role, showed a significant decrease. Our results demonstrate that upregulated PD-L1 inhibits CD8+T cells, causing immune escape of cancer cells. Therefore, our results showed that high KDM1A expression indicates worse clinical outcomes by inhibiting antitumoral immune activity. In a previous study, expression of KDM1A in breast cancer was reported for its association with cytotoxic T cell chemokines and PD-L1 and was consistent with our results [37,38]. In support of these results, we found high KDM1A expression to be significantly related to decreased CD8+T cells using TCGA data. High KDM1A expression was related to upregulated CTA expression, which is associated with cancer antigens in various types of malignancies. CTAs play an important role in tumorigenesis because upregulated CTAs are associated with progression and worse prognosis [39]. Our findings show that epigenetic reprograming by an epigenetic modification, KDM1A, plays an important role in mediating recruitment of CD8 + T cells to the tumor site.

Through analysis of functionally grouped networks, our study confirmed that high KDM1A expression is directly associated with AR signaling pathways. AR expression in breast cancer can vary from 53% to 93% depending on the subtype. The role of AR signaling depends on ER positivity and negativity in breast cancer, acting as a potential tumor suppressor with ER positivity but as a potential oncogene with ER negativity [40,41,42]. KDM1A is essential role for androgen-dependent gene transcription by demethylation of repressive marks such as histone H3 at lysine 9 (H3-K9) [43]. Previous studies identified breast development and metastasis pathways of androgen-androgen receptor/KDM1A target genes, and androgen-AR was identified as an oncogenic factor with epigenetic mechanisms involved in breast cancer carcinogenesis [44].

We investigated 175 anticancer drugs to which 50 breast cancer cell lines with high KDM1A expression were responsive by using pharmacogenomic screens from the GDSC database [26]. We identified six anticancer drugs that effectively reduce the growth of breast cancer cells with high KDM1A expression. Among them, alisertib, which showed a high negative correlation between high KDM1A expression and LN IC50, was identified as the most sensitive anticancer drug for breast cancer cell lines. Alisertib is currently being investigated in advanced solid tumors as an Aurora A kinase (AAK)-selective oral administration small molecular inhibitor. Multiple clinical trials have shown antitumor activity for the combination of alisertib with other therapies for advanced breast cancer [45,46]. AAK is an important member of the serine/threonine kinases involved in the regulation of cell mitosis. Indeed, the activity of AAK peaks during the transition from the G2 to M phase of the cell cycle [47]. Previous studies have found that KDM1A deficiency causes partial cell cycle arrest at G2/M phase [48]. Therefore, high KDM1A expression is associated with the growth of breast cancer cells and the effects of alisertib. Along with in vivo studies, alisertib-based clinical trials in breast cancer patients with high KDM1A expression are needed in the future.

This study has several limitations. First, since this study was retrospective, a decisive conclusion is difficult. Second, our study analyzed the oncogenic role of high KDM1A expression in breast cancer using a bioinformatics approach. Experimental analysis of the relationship between KDM1A and immune cells was not performed, and further in vitro and/or in vivo studies are necessary to identify the molecular mechanisms responsible. Third, the relationship between expression of KDM1A by intrinsic breast cancer subtypes and prognosis was not analyzed. Despite these limitations, our study revealed the functional mechanism of KDM1A in breast cancer through bioinformatics analysis using a large-scale database.

In summary, our study revealed that high KDM1A expression is significantly associated with downregulation of gene sets linked to CD8+T cells and decreased CD8+T cells. This implies that high KDM1A expression results in a worse prognosis in patients with breast cancer. We identified that alisertib effectively inhibits breast cancer cell lines with high KDM1A expression. This candidate drug has the potential to increase breast cancer survival in patients with high KDM1A expression and resistance to chemotherapy.

We believe that medical oncologists and researchers will be interested in the role of KDM1A in histone methylation and growth of breast cancer and that our results will facilitate further studies. In addition, our analytic workflow for KDM1A will contribute to designing future experimental studies and future drug development for patients with breast cancer.

## Figures and Tables

**Figure 1 jcm-10-01112-f001:**
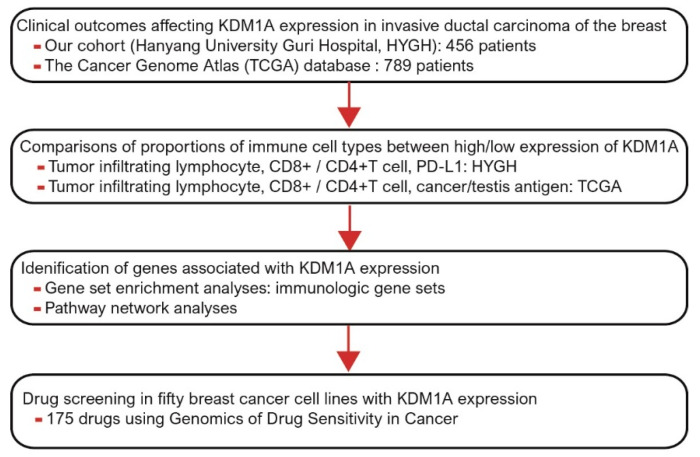
A schematic diagram depicting the plan of the study. KDM1A, Lysine-specific demethylase 1A; PD-L1, programmed death-ligand 1.

**Figure 2 jcm-10-01112-f002:**
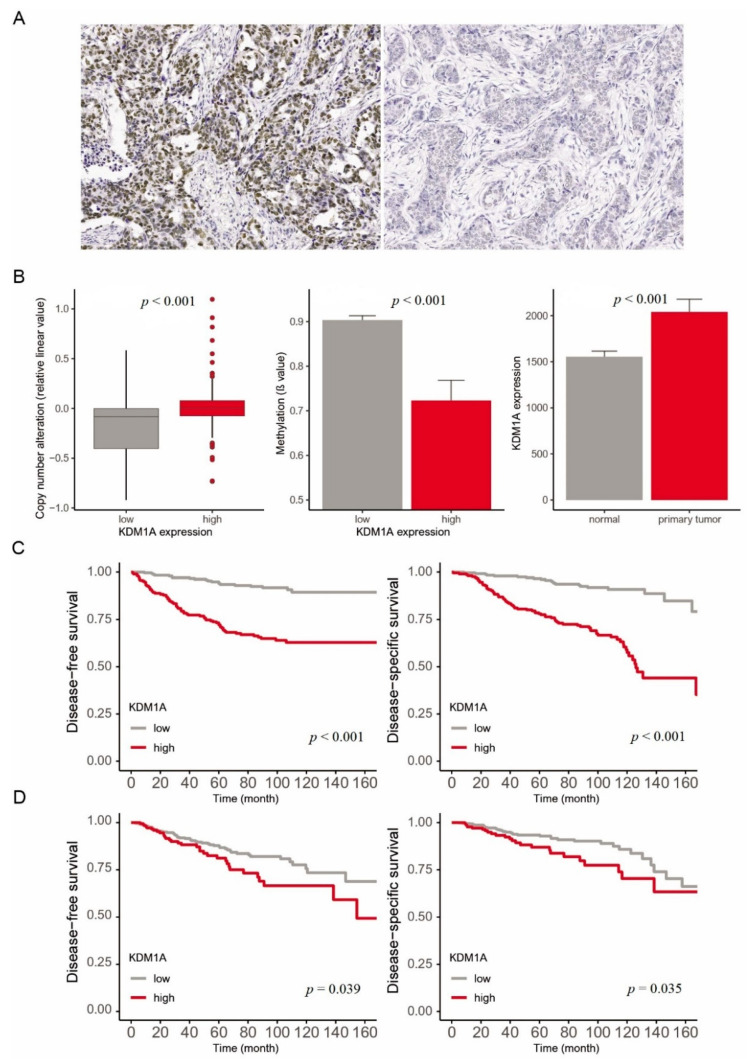
(**A**) HYGH: KDM1A expression showing positive (left) and negative (right) immunohistochemical staining in breast cancer (original magnification ×400). (**B**) TCGA: compared with normal tissue, high KDM1A expression was related to high copy number gain, hypomethylation, and primary tumors (all *p* < 0.001). (**C**) HYGH: high KDM1A expression was associated with poor disease-free survival and disease-specific survival in 456 patients (all *p* < 0.001). (**D**) TCGA: high KDM1A was associated with poor disease-free survival and disease-specific survival in 789 patients (*p* = 0.039 and 0.035, respectively).

**Figure 3 jcm-10-01112-f003:**
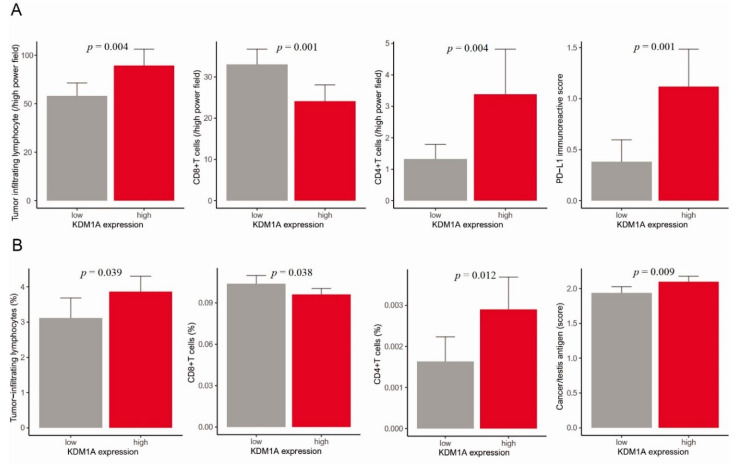
(**A**) HYGH: Bar plots of KDM1A and the following parameters: tumor-infiltrating lymphocytes, CD8+T cells, CD4+T cells and PD-L1 (*p* = 0.004, 0.001, 0.004 and 0.001, respectively) (error bars: standard errors of the mean). (**B**) TCGA: bar plots of KDM1A and the following parameters: tumor-infiltrating lymphocytes, CD8+T cells, CD4+T cells and cancer/testis antigen (score) (*p* = 0.039, 0.038, 0.012 and 0.009, respectively) (error bars: standard errors of the mean).

**Figure 4 jcm-10-01112-f004:**
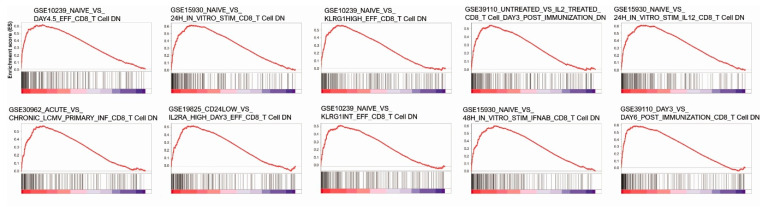
Gene set enrichment analyses of ten KDM1A-dependent immunologic gene sets revealed downregulation of CD8+ T cells.

**Figure 5 jcm-10-01112-f005:**
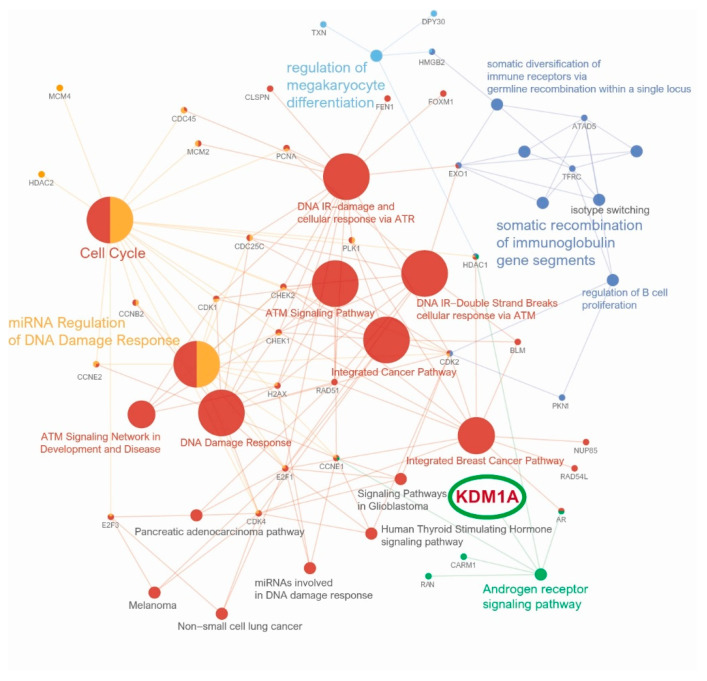
Network/pathway analyses: KDM1A is directly linked to the androgen receptor signaling pathway (green) and is indirectly linked to the immune response, DNA damage response, and cancer cell cycle pathway, among others.

**Figure 6 jcm-10-01112-f006:**
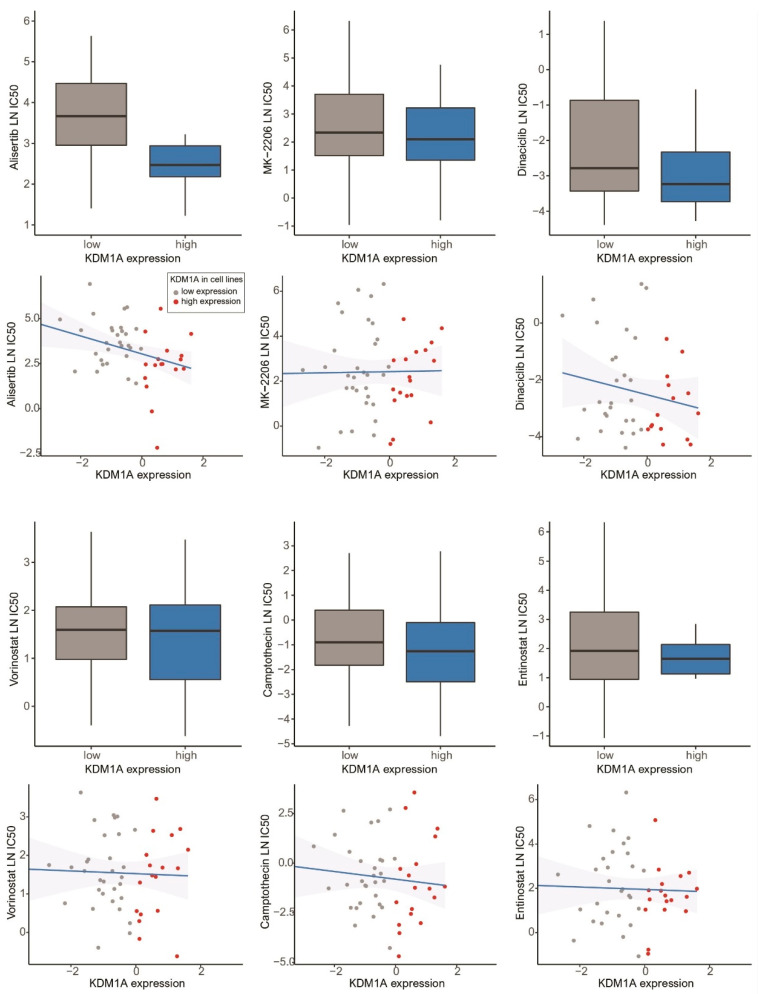
Bar plots and Pearson’s correlations showing the natural log half-maximal inhibitory concentration (LN IC50) of alisertib, MK-2206, dinaciclib, vorinostat, camptothecin, and entinostat, the most sensitive anticancer drugs in breast cancer cell lines with high KDM1A expression (error bars: standard errors of the mean).

**Table 1 jcm-10-01112-t001:** Clinicopathological parameters of KDM1A in 456 patients with invasive ductal carcinoma of the breast from HYGH.

Parameter	KDM1A Expression		*p*-Value
	Low (*n* = 246), *n* (%)	High (*n* = 210), *n* (%)	
Age (years)	49.6 ± 10.0	49.3 ± 9.9	0.791 ^1^
T stage			
1	122 (49.6%)	86 (41.0%)	0.122 ^2^
2	115 (46.7%)	111 (52.9%)	
3	9 (3.7%)	13 (6.2%)	
N stage			
0	137 (55.7%)	96 (45.7%)	0.001 ^2^
1	78 (31.7%)	57 (27.1%)	
2	24 (9.8%)	36 (17.1%)	
3	7 (2.8%)	21 (10.0%)	
Histological grade			
1	58 (23.6%)	29 (13.8%)	<0.001 ^2^
2	129 (52.4%)	87 (41.4%)	
3	59 (24.0%)	94 (44.8%)	
Lymphatic invasion			
Negative	140 (56.9%)	90 (42.9%)	0.004 ^3^
Positive	106 (43.1%)	120 (57.1%)	
Perineural invasion			
Negative	199 (80.9%)	163 (77.6%)	0.456 ^3^
Positive	47 (19.1%)	47 (22.4%)	
ER			
Negative	59 (24.0%)	70 (33.3%)	0.035 ^3^
Positive	187 (76.0%)	140 (66.7%)	
PR			
Negative	84 (34.1%)	95 (45.2%)	0.02 ^3^
Positive	162 (65.9%)	115 (54.8%)	
HER2			
Negative	203 (82.5%)	140 (66.7%)	<0.001 ^3^
Positive	43 (17.5%)	70 (33.3%)	
Ki-67 index	10.3 ± 13.5	9.8 ± 12.0	0.669 ^1^
P53 percentage	19.6 ± 31.4	37.2 ± 40.2	<0.001 ^1^

T or N stage, 8th edition; ER, estrogen receptor; PR, progesterone receptor; HER2, human epidermal growth factor receptor 2; KDM1A, lysine-specific demethylase 1A. ^1^ Student’s *t*-test ^2^ Linear-by-linear association ^3^ Chi-square test.

**Table 2 jcm-10-01112-t002:** Disease-free survival and disease-specific survival analyses according to KDM1A in 456 breast cancer patients from HYGH.

Disease-Free Survival	Univariate ^1^	Multivariate ^2^	HR	95% CI	
KDM1A (low vs. high)	<0.001	<0.001	3.241	1.969	5.336
Age (≤50 vs. >50)	0.768	0.11	1.389	0.928	2.078
T classification (1, 2 vs. 3, 4)	<0.001	<0.001	3.13	1.769	5.541
N classification (0, 1, 2 vs. 3)	<0.001	<0.001	2.617	1.556	4.4
Histological grade (1 vs. 2, 3)	0.001	0.022	2.337	1.128	4.842
Lymphovascular invasion (absence vs. presence)	<0.001	0.002	1.892	1.269	2.82
Disease-specific survival	Univariate ^1^	Multivariate ^2^	HR	95% CI	
KDM1A (low vs. high)	<0.001	<0.001	4.095	2.477	6.770
Age (≤50 vs. >50)	0.098	0.363	1.219	0.796	1.867
T classification (1, 2 vs. 3, 4)	<0.001	0.002	2.952	1.508	5.778
N classification (0, 1, 2 vs. 3)	<0.001	0.000	3.099	1.753	5.477
Histological grade (1 vs. 2, 3)	<0.001	0.043	2.377	1.027	5.505
Lymphovascular invasion (absence vs. presence)	<0.001	0.019	1.747	1.096	2.786

^1^ Log rank test; ^2^ Cox proportional hazard model.

## Data Availability

The data presented in this study can be available on request from the corresponding author. The data are not publicly available due to privacy.

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
