# Peer review of "High KDM1A Expression Associated with Decreased CD8+T Cells Reduces the Breast Cancer Survival Rate in Patients with Breast Cancer"

_jcm, 2021, doi:10.3390/jcm10051112_

Round 1

Reviewer 1 Report

Nice paper with detailed explanation and with many future insights into breast cancer treatment. I appreciate the authors mentioning the limitations of their study in the paper as well.

I suggest these minor changes to the article 

  1. Article title needs to be adjusted to make it more clear. The word "improves" and "Survival rate" in the title gives a positive signal to the readers (at least it sounded to me) that having high KDM1A improves breast cancer survival but the end result is a bad prognosis with decreased survival. I suggest adjusting the title with something else that can give clarity that high expression of the KDM1A has a worse prognosis for the breast cancer patients
  2. This sentence in the abstract results section "In silico flow cytometry results revealed low abundances of CD8+ T cells and those with high PD-L1 with KDM1A expression", needs to rewritten. I assume you are trying to mention that "Low abundances of CD8+ T cells and high PD-L1 expression in those with high KDM1A expression"

Author Response

We would like to extend our gratitude to you and the reviewers of the “Journal of clinical medicine” for taking the time and efforts to review our manuscript. Many of the valuable and constructive points you raised truly inspired the authors. After considering the reviewers’ comments, we revised the manuscript and have indicated the corrections and changes made with yellow highlights in the manuscript.

We now wish to submit the revised manuscript. The specific revisions and corrections made in response to the reviewers’ comments are as follows:

We have uploaded the marked file.

Reviewer Comments:
Reviewer 1
1. Article title needs to be adjusted to make it more clear. The word "improves" and "Survival rate" in the title gives a positive signal to the readers (at least it sounded to me) that having high KDM1A improves breast cancer survival but the end result is a bad prognosis with decreased survival. I suggest adjusting the title with something else that can give clarity that high expression of the KDM1A has a worse prognosis for the breast cancer patients

Answer:

We appreciate your valuable insight. The title of the journal refers to an increase in the survival rate of breast cancer cells due to a decrease in immune function. In other words, it represents a reduced survival rate for breast cancer patients. Your suggestion for a new title would reduce the confusion of our readers on the positivity and the negativity of our word choices. As you pointed out, the title has been revised to deliver more clear meaning and is indicated in the manuscript.

  1. This sentence in the abstract results section "In silico flow cytometry results revealed low abundances of CD8+ T cells and those with high PD-L1 with KDM1A expression", needs to rewritten. I assume you are trying to mention that "Low abundances of CD8+ T cells and high PD-L1 expression in those with high KDM1A expression"

Answer:

As recommended, we have modified the result in the abstract as follows and is indicated in the manuscript: low abundances of CD8+ T cells and high PD-L1 expression in those with high KDM1A expression.

Reviewer 2 Report

High KDM1A Expression Associated with Decreased CD8+ T Cells Improves the Breast Cancer Cell Survival Rate in Patients with Breast Cancer.

This manuscript by Kim et al. reports a strong correlation between high expression of the histone demethylase KDM1A (also known as lysine-specific histone demethylase) and poor survival rates and immune response in breast cancer patients. This is a correlation study where only bioinformatic data has been used to establish an association between KDM1A and various clinicopathological breast cancer features. Although there were valuable concepts and correlations identified throughout the manuscript, they were not clearly articulated or justified. Nonetheless the results from this paper do provide valuable insights about KDM1 and lay a foundation for future studies into the role of KDM1A in breast tumorigenesis.

Major Concerns

  1. The title and many statements/conclusions in the manuscript are over-reaching and inaccurate; for example, “High KDM1 ….improves breast cancer cell survival rate…”. The title should be revised to state that high KDM1 correlates with poor survival in breast cancer patients which is a more accurate statement of what was found. The manuscript could be significantly improved with the inclusion of mechanistic data and analyses.
  2. As no mechanistic experiments were conducted in this study, the conclusions are also over-reaching since many genes are upregulated in breast cancer but this does not establish a “cause and effect” outcome. The authors need to conduct mechanistic studies in cell culture or mouse models to demonstrate the KDM1 plays a definitive and functional role in breast cancer and to justify their statement in the discussion that “to their knowledge, this is the first large-scale study to show the functional mechanism of KDM1A in breast cancer…”
  3. The authors never stated what the rationale was for correlating KDM1 with immune markers, CD8+ etc
  4. The authors never stated how the optimal cut-off for high vs low KDM1 determined. This was not explained and needs to be clearly stated/explained and defined. The cut-off for TCGA was also not defined.
  5. Manuscript could be improved if KDM1 expression was correlated with different breast cancer subtypes.
  6. The importance of the androgen receptor signaling network and the link to histone demethylation and KDM1 was not explained.
  7. The author provided no rationale for looking at anti-cancer drugs and correlating with KDM1 expression. This aspect of the study needs a rationale and mechanistic studies in in cell lines or mouse models and xenografts. Many other genes and proteins are altered by anti-cancer drugs, and thus one cannot simply extrapolate and make strong conclusions as done in this manuscript.

Author Response

We would like to extend our gratitude to you and the reviewers of the “Journal of clinical medicine” for taking the time and efforts to review our manuscript. Many of the valuable and constructive points you raised truly inspired the authors. After considering the reviewers’ comments, we revised the manuscript and have indicated the corrections and changes made with yellow highlights in the manuscript.

We now wish to submit the revised manuscript. The specific revisions and corrections made in response to the reviewers’ comments are as follows:

We have uploaded the marked file.

  1. The title and many statements/conclusions in the manuscript are over-reaching and inaccurate; for example, “High KDM1 ….improves breast cancer cell survival rate…”. The title should be revised to state that high KDM1 correlates with poor survival in breast cancer patients which is a more accurate statement of what was found. The manuscript could be significantly improved with the inclusion of mechanistic data and analyses.

Answer:

We appreciate your valuable insight. The title of the journal refers to an increase in the survival rate of breast cancer cells due to a decrease in immune function. In other words, it represents a reduced survival rate for breast cancer patients. Your suggestion for a new title would reduce the confusion of our readers on the positivity and the negativity of our word choices. As you pointed out, the title has been revised to deliver more clear meaning and is indicated in the manuscript.

  1. As no mechanistic experiments were conducted in this study, the conclusions are also over-reaching since many genes are upregulated in breast cancer but this does not establish a “cause and effect” outcome. The authors need to conduct mechanistic studies in cell culture or mouse models to demonstrate the KDM1 plays a definitive and functional role in breast cancer and to justify their statement in the discussion that “to their knowledge, this is the first large-scale study to show the functional mechanism of KDM1A in breast cancer…”

Answer:

We appreciate the good point raised by the reviewer. We also understand some of the limitations in our study because our research is not based on experimental data. In order to overcome such limitations, we have obtained the results with a bioinformatics analysis method with a large TCGA database. We plan to conduct an experimental analysis through these concepts in future research. As recommended, we added revised sentences in “discussion” section as follows :

Despite these limitations, our study revealed the functional mechanism of KDM1A in breast cancer through bioinformatics analysis using a large-scale database.

  1. The authors never stated what the rationale was for correlating KDM1 with immune markers, CD8+ etc

Answer:

Epigenetic dysregulation plays a critical role in silencing expression of certain effector T cell chemokines, which may lead to inefficient recognition and elimination of cancer cells by the host immune system.

As recommended, we added new sentences in “discussion” section as follows :

Increased TIL is known to be a good prognostic factor for cancer because lymphocytes play an important role in anti-cancer immunity. Among lymphocytes, CD8+T cells play critical role in anti-cancer immunity. Our results showed that high expression of KDM1A was associated with an increase in TIL, while CD8+T cells, which play an important role, showed a significant decrease.

In a previous study, expression of KDM1A in breast cancer was reported for its association with cytotoxic T cell chemokines and PD-L1 and was consistent with our results.”

Our findings show that epigenetic reprograming by an epigenetic modification, KDM1A, plays an important role in mediating recruitment of CD8 + T cells to the tumor site.”

  1. The authors never stated how the optimal cut-off for high vs low KDM1 determined. This was not explained and needs to be clearly stated/explained and defined. The cut-off for TCGA was also not defined

Answer:

On basis of the cancer specific death events in the HYGH cohort, the immunoreactive score (IRS = 5) was the most sensitive and specific value through the receiver operating characteristic (ROC) curve analysis for the IRS of KDM1A.

As recommended, we added new sentences in “materials and methods” section as follows:

“To determine the optimal cutoff values of KDM1A, receiver operating characteristic (ROC) curves plotting sensitivity versus 1 – specificity were used. The cutoff value calculated by the ROC was used to evaluate the relationship between cancer specific death events and KDM1A expression.”

On basis of the cancer specific death events in the TCGA database, the values of KDM1A were divided into low and high using the most sensitive and specific value in the receiver operating characteristic (ROC) curve analysis.

The cutoff value was 2383.075 (log2-transformed score 11.543)

As recommended by the reviewer, we added the content in the “Materials and method” section as follows:

The RNA seq from TCGA was calculated, and KDM1A expression was determined as either low (log2 -transformed scores < 11.543) or high (log2 -transformed scores > 11.543)

  1. Manuscript could be improved if KDM1 expression was correlated with different breast cancer subtypes.

We appreciate your valuable insight.

As the reviewers have suggested, our study may show better results when analyzed as a breast cancer subtype. We confirmed the expression level of KDM1A according to each ER, PR, and HER2, but did not confirm the relationship between the expression level or survival rate according to the subtype. In the limitation section, we explained that we were not able to perform analyzes by subtype of breast cancer.

We conducted this study focusing on the clinical significance of the overall breast cancer of the high expression of KDM1A and the functional mechanisms related to immune cells. And, our study tried to find evidence before conducting subsequent studies based on experimental studies. We plan to analyze each subtype of breast cancer in a follow-up study.

  1. The importance of the androgen receptor signaling network and the link to histone demethylation and KDM1 was not explained.

We appreciate the good point. KDM1A interacts with androgen receptor in vitro and in vivo, and stimulates androgen-receptor-dependent transcription.

We added new descriptions in “discussion” section as follows:

KDM1A plays an essential role for androgendependent gene transcription by demethylation of repressive marks such as histone H3 at lysine 9 (H3-K9).”

  1. 7. The author provided no rationale for looking at anti-cancer drugs and correlating with KDM1 expression. This aspect of the study needs a rationale and mechanistic studies in in cell lines or mouse models and xenografts. Many other genes and proteins are altered by anti-cancer drugs, and thus one cannot simply extrapolate and make strong conclusions as done in this manuscript.

Answer:

We appreciate the good point.

We understand some of the limitations in our study because our research is not based on experimental data. Therefore, we have described in the conclusion as follows: “We believe that medical oncologists and researchers will be interested in the role of KDM1A in histone methylation and growth of breast cancer and that our results will facilitate further studies. In addition, our analytic workflow for KDM1A will contribute to designing future experimental studies and future drug development for patients with breast cancer.”

As our research is not based on experimental data, we plan to find the significance and target drugs of KDM1A as a breast cancer biomarker and confirm the results of these studies through experimental methods in subsequent studies. This study was conducted to find evidence for subsequent research through experimental methods.
